# Adsorption and pH Values Determine the Distribution of Cadmium in Terrestrial and Marine Soils in the Nansha Area, Pearl River Delta

**DOI:** 10.3390/ijerph19020793

**Published:** 2022-01-11

**Authors:** Fangting Wang, Ke Bao, Changsheng Huang, Xinwen Zhao, Wenjing Han, Zhibin Yin

**Affiliations:** 1Wuhan Geological Survey Center, China Geological Survey, Wuhan 430205, China; ftwang1991@cug.edu.cn (F.W.); zhaoxinwen@mail.cgs.gov.cn (X.Z.); 2Safety Center for River and Lake Protection, Construction and Operation, Changjiang Water Resources Commission of the Ministry of Water Resources, Wuhan 430015, China; kebao@cug.edu.cn; 3Geological Survey Research Institute, China University of Geosciences, Wuhan 430074, China; hanwenjing@cug.edu.cn (W.H.); zhibinYY@cug.edu.cn (Z.Y.)

**Keywords:** cadmium contamination, soil pH, isothermal adsorption experiment, adsorption kinetic experiment, longitudinal migration, the Pearl River Delta

## Abstract

Cadmium is a toxic element with a half-life of several decades, which can accumulate in the human body by entering the food chain and seriously harm health. The cadmium adsorption and desorption processes in the soil directly affect the migration, transformation, bioavailability, and ecotoxicity of this element in soil-plant systems. Coastal zones are located in the transitional zone between land and sea, and large amounts of terrigenous material input have important environmental effects on this ecosystem. The pH, hydrodynamic conditions, soil organic matter (SOM), and other factors defining the sea-land interaction within the sedimentary environment are significantly different from those defining land facies. In order to study the key factors affecting cadmium adsorption in soils at the sea-land interface in the Nansha area of the Pearl River Delta, a test was conducted on a column of undisturbed soil. The results showed that the adsorption constant *K_F_* and the Cd^2+^ adsorption capacity of marine soils were higher than those of terrestrial soils. However, the saturation adsorption of cadmium in terrestrial sediments was higher than in marine sediments. Soil pH was an important factor affecting cadmium adsorption capacity in both terrestrial and ma-rine sediments. Neutral and alkaline topsoil conditions inhibited the vertical migration of cadmium, while the acidic environment favored it. The higher the clay and SOM were, the stronger the Cd^2+^ adsorption capacity of the soil was. These findings suggest that the distribution of cadmium in marine and continental sedimentary soils is not only related to adsorption, but also to the physical and chemical processes occurring in different sedimentary environments.

## 1. Introduction

Cadmium (Cd) is a carcinogen that can be accumulated in the body by entering the food chain and seriously endanger human health [1]. Excessive importation of this heavy metal into the soil has attracted worldwide attention because of its high toxicity, persistence, and bio-enrichment capacity [2,3]. Survey results show that 13.86% of China’s grain production is affected by inorganic pollutants present in soils, among which cadmium ranks as the most polluting metal, accounting for 7.75% [4]. Multi-target regional geochemical surveys conducted in China show that cadmium pollution in land areas around the Pearl River Delta is increasing. Major factors contributing to the increase are: wastewater irrigation, atmospheric precipitation, stacking and leaching of solid waste in industrial and mining enterprises, and the use of agricultural supplies [5]. Overall, the environmental quality of the soil in the Pearl River Delta is poor. Most of the soils in the area, covering an area of 9298.3 km^2^, are classified as grade III or inferior to grade III. The main toxic elements detected are Cd, Hg, As, Cu, and F [6]. Due to its stable chemical properties, a small part of the cadmium entering the soil is absorbed by the soil medium, while the rest is transferred and transformed in the soil environment, resulting in soil cadmium pollution. Plants absorb cadmium from the contaminated soil and accumulate large amounts of this element, which then enters the food chain once the edible parts of plants are ingested by various animal species [7]. Cadmium-contaminated soils will eventually produce cadmium-contaminated grains, posing a serious threat to human health [8]. Therefore, finding viable solutions to reduce the health risk posed by cadmium pollution has become a research priority in recent years.

The cadmium adsorption and desorption processes occurring in the soil are key processes that control cadmium concentration in soil solutions, which directly affect the migration, transformation, bioavailability, and ecotoxicity of this element in soil-plant systems [9], ultimately affecting the quality of agricultural products and of the environment. Therefore, it is of great practical significance to study the adsorption and desorption characteristics of cadmium in order to understand and control its behavior in the soil-plant system and to predict its environmental effects [10]. The cadmium adsorption behavior in the soil varies with soil type and is affected by soil properties such as pH [11,12], soil organic matter (SOM) [13], ion exchange capacity, and clay content [14]. Cadmium exists in nature as a divalent form, with stable chemical properties, and it is easily soluble in acid solutions. As the surface soil becomes polluted with heavy metals, cadmium is precipitated out under the leaching action of acid rainfall and it gradually sinks and migrates, which may pollute phreatic water aquifers in serious cases. The Pearl River Delta region is characterized by permanent agricultural activities and abundant rainfall [15], which is a distinctive feature of the subtropical monsoon climate [16]. Acid rain is a widespread phenomenon in the area, with rainfall pH values ranging between 4.0 and 5.6; surface soil pH values are usually low, thus affecting cadmium migration, transformation, and bioavailability. On the other hand, coastal zones represent an area where important interactions occur between land, ocean, and atmosphere. These zones are vulnerable to environmental change, and they are highly economically developed. At the land-sea interface, various physical, chemical, and biological factors are subjected to dramatic variations, making this area particularly fragile. It is very sensitive to the fluctuations caused by various natural processes and by reactions in response to human activities [17]. Moreover, the large amounts of land-originated terrigenous input have important environmental effects on the entire coastal ecosystem [18]. Geomorphology in the Pearl River Delta is determined by the sea-land interaction, which also affects the geochemical characteristics of the quaternary soils present in the area [19,20]. The pH, oxidation-reduction potential (Eh), hydrodynamic conditions, SOM, and other factors pertaining the sedimentary environments of the sea-land interface are significantly different from those of continental facies, which usually constitute geochemical barriers to the elements that migrate stably in them, ultimately leading to the precipitation and enrichment of elements in the sea-land transition zone [21].

Several studies have investigated the effects of soil properties on the adsorption-desorption, transport, and transformation of heavy metals [22,23,24]. However, these studies primarily focused on surface soils of cultivated lands, and often ignored the soil properties of different sedimentary environments, both terrestrial and marine. In order to prevent and reduce the environmental damage caused by cadmium in the sea-land interaction zone of the Pearl River Delta, it is urgent to study the mechanisms of cadmium migration and transformation in soils within different sedimentary environments. Various studies have investigated the long-term equilibrium effect of metals in soil, but have not paid enough attention to the influence of important soil properties, such as pH [25,26]. Consequently, the interaction between the soil’s depositional environment and soil properties during metal adsorption is poorly understood.

This paper is one of a series of studies on the effects of soil properties on heavy metal adsorption. In Wang et al. (2019) [27], multivariate statistical and spatial analysis methods were used to analyze the factors affecting the accumulation of cadmium in sedimentary soils at the sea-land interaction zone in the Pearl River Delta. Here, we further extended the model of cadmium adsorption in soils within marine and continental inter-deposition environments considering important additional environmental parameters. In terms of global geography, China’s coastal areas lie between the largest continent and the largest ocean, and China’s marginal sea is a natural laboratory for studying the interaction between sea and land [28]. In this study, the soil of both terrestrial and marine sedimentary environments in the Nansha area of the Pearl River Delta and its cadmium adsorption characteristics were studied using isothermal adsorption methods. Through the measurement of external environmental parameters (pH value, temperature, time, etc.), the cadmium adsorption behavior was analyzed in terms of isothermal and kinetic adsorption; additionally, the adsorption characteristics and influencing factors at play within the sea-land transition zone were explored.

## 2. Materials and Methods

### 2.1. Study Area

The Nansha district connects the Pearl River basin to the ocean, and covers a total area of approximately 803 km^2^ (Figure 1). The region is characterized by a southern subtropical monsoon climate, the annual average temperature is 22.6 °C, and the annual average rainfall is 1673.1 mm. This region is highly affected by thunderstorms which occur 72.2 days per year [29]. The landforms in Nansha are hilly, marine–terrigenous plains, and tidal flats. The low hills are mainly distributed around Huangshanlu Mountain and the highest point in the region is 295.3 m above sea level. The marine–terrigenous plains are distributed in Huangge and Hengli; the upper part is silt or silty soil (sand) and the lower part is river-deposited sand. The tidal flats are mainly distributed in the southeastern area of Wanqingsha, Longxue Island, and the Xinken coast. The terrain is zonal, parallel to the coast, broad, and gentle.

### 2.2. Soil Sampling and Chemical Analysis

Two types of soils were collected in Wanqingsha Town, Nansha District, to be used in laboratory adsorption experiments to study the adsorption characteristics of soils with different origins. Soil sample 1 belonged to the continental sediment layer; it was identified as silted clay and was collected at a depth ranging from 0 to 0.5 m. Soil sample 2 belonged to the marine sediment layer, it was identified as sandy silt and was collected at a depth ranging from 1 to 1.5 m. All soil samples were air-dried, mashed with a 2 mm screen, and sealed separately in airtight polyethylene bags.

A total of 18 groups of deep soil profiles (0~200 cm) were collected in the Nansha District, Guangzhou (Figure 1C). The sampling sites were mainly located in the thicker soil layer of gardens. Care was taken to avoid sections with obvious point source pollution, newly transported and accumulated soil, and soil occupied by garbage and ridge fields; sampling was conducted at least 100 m away from main highways and railways. The deep soil was sampled by hand drill, and the soil samples were collected in each section at a depth between 0 and 2 m. They were divided into five groups (0~20, 20~50, 50~90, 90~140, and 140~200 cm), A total of 90 groups of soil samples were sealed in airtight polyethylene bags.

All the soil samples were taken back to the laboratory and the physical and chemical indexes were determined. Soil pH was determined by a pH meter (Ion selective electrode method) with an accuracy of 0.01 [30]. The SOM content and cation exchange capacity (CEC) were determined using the potassium dichromate volumetric method and ammonium exchange method referring to standards [30,31]. The total cadmium content in soil samples was determined by an inductively-coupled plasma mass spectrometer with a detection limit of 0.02 μg/g [32]. According to the International Classification of Soil Texture, soil particles are classified into the following types [33]: clayey (<0.002 mm), silty (0.002~0.02 mm), and sandy (0.02~2 mm).

### 2.3. Batch Experiments

The main reagents used in the experiments were pH standard buffer solution (4.01, 6.86, 9.18), hydrochloric acid, and the national standard cadmium solution (product number: GSBG62040-90, concentration: 1000 μg/mL, medium: 10%HCl). The chemical reagents used were high-purity analytical reagents. Deionized water (18.2 MΩ·cm) was prepared with a Heal Force NW ultrapure water system and used in all experiments.

#### 2.3.1. Isothermal Adsorption Experiment

A series of standard solutions containing cadmium concentrations of 0.1, 0.5, 1, 5, 10, 50, 100, 150, 200, 250, 300, 350, 400, and 450 mg/L were prepared, with 0.01 mol/L NaCl solution as supporting electrolyte solution. The basis for setting the soil-liquid ratio was as follows: the volume mass (bulk density) of the experimental soil was about 1.5 g/cm^3^, and considering that the 0~20 cm topsoil layer near was the main layer affected by acid rain, and the weight of soil column per 1 cm^2^ unit area was 30 g. According to meteorological data, the annual rainfall in Guangzhou is about 1646.9 mm. Therefore, the soil-liquid ratio used in the experiment was 30 g:(1 g/cm^3^ × 1646.9 mm × 1 cm^2^) ≈ 1:5.5. A quantity of 8 ± 0.001 g of soil sample, and 44 mL of a series of standard cadmium solutions were added to a 50 mL plastic centrifuge tube. The mixtures were evenly mixed and shaken at a constant temperature of 25 ± 1 °C for 24 h in a thermostatic oscillator to obtain full adsorption (two sets of parallel samples for each cadmium solution, and a total of 28 groups of mixed samples were prepared), and then they were centrifuged at 4030 r/min for 20 min. The supernatant was removed to analyze Ph (PHS-3C calibrated daily) and conductivity (DDS-307A). The concentration of cadmium in the supernatant was measured with an atomic absorption spectrophotometer (ZCA-1000SF8).

The adsorption effect is usually defined by the adsorption capacity and adsorption rate: the adsorption capacity *S* and adsorption rate *η* of soil to cadmium are calculated as follows [34]:*S* = (*C*_0_ − *C_e_*)*V*/*m*(1)
*η*(%) = (*C*_0_ − *C_e_*) × 100/*C*_0_(2)
where, *S* is the adsorption capacity of soil to cadmium, mg/kg; *η* is the adsorption rate of soil to cadmium, %; *C*_0_ is the initial concentration of cadmium in the solution, mg/L; *C_e_* is the concentration of cadmium in the equilibrium solution, mg/L; *V* is the volume of the equilibrium solution, mL; *m* is the mass of the soil, g.

#### 2.3.2. Adsorption Kinetic Experiment

Using 0.01 mol/L NaCl as the supporting electrolyte solution and 0.1 mol/L HCl to adjust pH to 3.5, 4.5, 5.5, and 9.5, a standard solution with a cadmium concentration of 100 mg/L at different pH values was prepared. A quantity of 8 ± 0.001 g of soil sample and 44 mL of standard cadmium solution were added to a 50 mL centrifuge tube in that order, based on the soil-liquid ratio of 1:5.5. The mixtures were evenly mixed and shaken at a constant temperature of 25 ± 1 °C in a thermostatic oscillator for 2 min, 5 min, 10 min, 15 min, 20 min, 25 min, 30 min, 1 h, 2.5 h, 4 h, 6 h, 8 h, 10 h, 12 h, and 24 h respectively (two sets of parallel samples for each time interval, and a total of 150 groups of mixed samples were taken). Each set of tubes was placed in a 5030 r/min centrifuge for 5 min immediately after the oscillation phase. The supernatant was removed to analyze pH (PHS-3C calibrated daily) and conductivity (DDS-307A). The concentration of cadmium in the supernatant was measured with an atomic absorption spectrophotometer. Finally, the adsorption capacity S of soil to cadmium was calculated based on the initial equilibrium concentration of cadmium in the solution.

### 2.4. Analysis

IBM SPSS Statistics 22.0 software (International Business Machines Corporation, Armonk, New York) was used for correlation analysis, and Origin 2018 software for experimental data sorting and analysis. The parameters of the isothermal adsorption and adsorption kinetics models were fitted using the results of the respective experiments.

The results of heavy metal adsorption analyses are usually expressed by isothermal adsorption curves. For different reactions, there are specific fitting equations that can describe adsorption behavior. By comparing various competitive adsorption equations, the fitting equations which are more similar to the experimental results are determined to predict the results. Isothermal adsorption equations with wide applicability and simple formulas include the Henry, Langmiur, and Freundlich equations, which are described as follows [34]:Henry equation: *S* = *K_d_C_e_*(3)
Freundlich equation: *S* = *K_F_C_e_*^1/*n*^(4)
Langmiur equation: *S* = *K_L_S_m_C_e_*/(1 + *K_L_C_e_*)(5)
where, *S* is the adsorption capacity, mg/kg; *K_d_* is the solid/liquid distribution coefficient, L·kg^−1^; *C_e_* is the concentration of equilibrium solution, mg/L; *K_F_* and 1/*n* are isothermal adsorption constants, *K_L_* is the value of soil adsorption affinity, and *S_m_* is the saturation adsorption, mg/kg.

Adsorption kinetics models are mainly used to describe adsorption rates, reactions at a stable temperature, relationships between adsorption substances, and time, all of which collectively describe the mechanisms of heavy metal adsorption by soils. A variety of dynamics equations were established and the fitting equations more similar to the experimental results were determined to predict the results. Dynamics model equations with wide applicability and simple formulas include the Elvoich equation, the double constant equation, and the pseudo-first-order dynamics equation. These are described as follows:Elvoich equation: *S_t_* = *m* + *n*ln*t*(6)
Biconstant equation: ln*S_t_* = *m* + *n*ln*t*(7)
Quasi first order kinetic equation: ln(*S_m_* − *S_t_*) = ln*S_m_* − (*K*_1_*t*)/2.303(8)
where, *S_t_* is the solute adsorption capacity of soil in time *t*, mg/kg; *m* and *n* are constants; *K*_1_ is the kinetic parameter; and *S_m_* is the saturation adsorption, mg/kg.

## 3. Results and Discussion

### 3.1. Physical and Chemical Properties of Soils in Two Sedimentary Environments

The land morphology of the sampling site is delta plain, and this land is primarily used for banana plantations. The soil in the first terrestrial sedimentary environment (Soil 1) was a yellowish silty clay with a 35% clay ratio, pH value of 6.6, weak acidity, 1.8% SOM content, and 0.52 mg/kg cadmium content. The soil in the marine sedimentary environment (Soil 2) was gray-brown sandy silt, moist, with a soft to fluid plasticity, 64% of sand proportion, an 8.08 pH value (alkaline), 1.58% SOM content, and 0.48 mg/kg cadmium content (Table 1). The cadmium background concentration in Guangdong Province soils was 0.056 mg/kg, while it was double in the Pearl River Delta, reaching 0.11 mg/kg [35]. The cadmium content measured in the two sedimentary soils in the study area was significantly higher than the background cadmium value of both Guangdong Province and Pearl River Delta.

The physical and chemical conditions of environmental media changed dramatically, while the activities of heavy metal elements in sediments changed correspondingly under the strong interaction between land and sea. Dou Lei et al. [36] summarized the elements content characteristics of marine and continental sedimentary environments based on the results of multi-objective geochemical investigation, and found that most elements content in marine sedimentary environments is higher than that in terrestrial sedimentary environments.

### 3.2. Effects of Initial Concentrations on Cadmium Adsorption in the Two Soils

During the isothermal adsorption experiment, 8 g of each soil sample was evenly mixed with 44 mL of cadmium solutions with different concentrations, and the effects of the soil on the adsorption capacity of cadmium were studied while vibration and a constant temperature of 25 ± 1 °C were maintained for 24 h. As shown in Figure 2A, the adsorption capacity of Cd^2+^ increased in all samples with the increase of the initial cadmium concentration, and the variation can be divided into three stages (Figure 2B): (1) when the initial concentration was between 0 and 50 mg/L, the adsorption ratio of Cd^2+^ on soil increased significantly, (2) when it was between 50 and 200 mg/L, the adsorption ratio of Cd^2+^ was high, still greater than 95%, and (3) when it was more than 200 mg/L, the adsorption ratio decreased gradually.

In Figure 3, the Henry, Freundlich, and Langmuir models are shown, respectively. These models were used to fit the adsorption of cadmium by the two soils and to establish the isotherm adsorption equation. According to the correlation coefficient analysis, the Freundlich equation has a better fitting for the cadmium adsorption process. The adsorption parameters of the two soils obtained through the three isothermal adsorption models are shown in Table 2.

Houng et al. (1998) [37] observed that the *K_F_* constant in the Freundlich model is an important parameter to evaluate the adsorption capacity of heavy metals during soil adsorption processes, and the value of *K_F_* is proportional to the capacity of the soil to adsorb heavy metal ions, but independent of saturation adsorption. Table 2 shows that the cadmium isothermal adsorption constant *K_F_* of soil 2 in the study area was significantly higher than that of soil 1, measuring 558.21 and 363.04, respectively. The higher the *K_F_* value is, the stronger the soil adsorption capacity for Cd^2+^ [38]. Therefore, the adsorption capacity of soil 2 was stronger than that of soil 1. This is because soil 1 is a neutral soil while soil 2 is alkaline, and the amount of negative charge on the colloidal surface of alkaline soils is generally greater, resulting in an enhanced electrostatic action of the soil. This, in turn leads to the enhancement of the obligate adsorption capacity of the colloidal surface to metal ions [39]. Furthermore, the SOM content of soil 1 was higher than that of soil 2, and the soil particles with more SOM content combine with inorganic colloids to form new complex colloids, thus weakening the adsorption capacity of soil particles to heavy metals. Nevertheless, the saturation adsorption of cadmium in soil 2 was smaller than that in soil 1, measuring 2228.25 mg/kg and 2419.83 mg/kg, respectively. This may be due to the clay content present in the soil. The specific surface area of clay particles is large, usually negatively charged, and has a high cation exchange capacity. Under the action of static force, positively charged cations will adsorb on the surface of clay minerals [40]. Therefore, it is believed that the higher the content of physical clay particles in the soil, the stronger the soil’s ability to adsorb Cd^2+^. The clay particle content in terrestrial sediments in the study area was high, accounting for 35% of the total, the cation exchange capacity is 15.2 cmol/kg, and the soil had a strong cadmium adsorption capacity. Conversely, the marine sediment soil has a high sand content (64%), a cation exchange capacity of 10.5 cmol/kg, and a weak soil adsorption capacity, indicating that clay minerals play an important role during the process of cadmium adsorption.

The solid-liquid phase equilibrium controls the mobility and toxicity of metals in soils. The higher the *K_d_* value is, the easier it is for Cd^2+^ to be fixed in the soil solution through adsorption; the lower the *K_d_* value is, the more Cd^2+^ exists in the soil solution [41]. The variation of the solid-liquid distribution coefficient *K_d_* with the initial solution concentration is shown in Figure 4. The variation trend of *K_d_* seen in soils from different sedimentary environments was basically the same, showing an increase first, then a decrease, and finally a stabilization with the increase of the initial concentration. Specifically, in this study, when the initial cadmium concentration of soil 1 was 0–10 mg/L, the *K_d_* value increased, reaching the maximum peak. At a concentration of 10–100 mg/L, the value decreased rapidly until it stabilized. When the initial concentration of soil 2 was 0–50 mg/L, the *K_d_* value increased reaching the maximum peak. At a concentration of 50–150 mg/L, the value decreased rapidly until it stabilized, indicating the soil’s ability to hold Cd^2+^ by reaching adsorption equilibrium. Considering that the stability value of the soil’s *K_d_* does not change significantly in different sedimentary environments—but the peak value of *K_d_* changes significantly—the latter was chosen to characterize the soil’s ability to retain Cd^2+^. By comparing the *K_d_* peak values of the two soil types, it emerged that the *K_d_* peak values of soil 2 were significantly higher than those of soil 1, indicating that Cd^2+^ in the soil 2 solution was more easily fixated by adsorption, and its adsorption capacity was stronger. This process depends on the amount of cation exchange between the two soils. Generally, a higher total metal concentration results in a lower proportion of metals being adsorbed because potential adsorption sites are filled in order of decreasing affinity. The more a soil becomes “saturated” with cations, the lower the metal affinity of the remaining sites. We would then expect the resulting *K_d_* values to decrease with the increasing of total metal levels [41].

### 3.3. Effect of Adsorption Time on Cadmium Adsorption in the Two Soils

Adsorption kinetics express the relationship between solution concentration and reaction duration under fixed temperature conditions. In this study, 8 g of each soil sample was evenly mixed with 44 mL Cd^2+^ solution with a concentration of 100 mg/L. By controlling the temperature (*t* = 25 ± 1 °C), it was possible to analyze the cadmium adsorption capacity of the soil at different times. As shown in Figure 5A, at the beginning of the adsorption process, the soil adsorbed Cd^2+^ combined with soil colloids by simple Coulomb force through the double potential layer (non-mandatory adsorption). The adsorption rate was fast and close to saturation, and the adsorption capacity reached the peak. Desorption occurred between one and 10 min, resulting in the increase of equilibrium concentration, and the soil adsorption capacity of cadmium decreased significantly from the maximum to the minimum value. After 10 min, the adsorption of Cd^2+^ changed into mandatory adsorption, and the adsorption rate was slow. With the increase of oscillation time, the equilibrium concentration gradually decreased, and the cadmium adsorption capacity on the soil gradually increased and tended to stabilize, reaching the equilibrium adsorption. By comparing the relationship curves of soil adsorption capacity and oscillation time in the two different sedimentary environments, it was observed that when the concentration of Cd^2+^ in aqueous solution was 100 mg/L, the initial maximum adsorption capacity of soil 1 and soil 2 (t = 1 min) was 529.65 and 539.33 mg/kg, respectively, and the equilibrium adsorption capacity (t = 24 h) was about 538 and 547 mg/kg, respectively. The initial maximum adsorption capacity and equilibrium adsorption capacity of soil 2 were both greater than those of soil 1.

The relationship between the adsorption capacity and time is characterized by the adsorption rate, which refers to the capacity of cadmium to be adsorbed by soil per unit mass in unit time [42], i.e., *V* = *S*/*t*. The relationship between the adsorption rate and adsorption time of marine and terrestrial soils is shown in Figure 5B. At the beginning of the shock experiment (*t* = 1 min), the adsorption rate was the fastest; then, with the increase in adsorption time, the soil adsorption rate slowed. At 2.5 h, the adsorption rate in the two soils was 3.6 mg/(kg·min), and after 24 h the rate was the slowest, at 0.4 mg/(kg·min), and in a state of equilibrium adsorption.

The biconstant equation, Elovich equation, and a first-order quasi-linear equation fitting were used to simulate the dynamic process of cadmium adsorption by terrestrial and marine soils in the Nansha area. The adsorption kinetic model was established based on the relationship between the amount of cadmium adsorbed and the adsorption time of the two soils. Both the Elovich equation (Figure 6A) and the biconstant equation (Figure 6B) could adequately describe the adsorption process, while the first order quasi-linear equation fitting was poor. However, a similar exponential asymptote equation could be used to fit the dynamic process of cadmium adsorption (Figure 6C), one of which provided better results than the other two equations. The dynamics fitting degree of the two soils was similar. Based on the exponential asymptote equation, when the ratio of soil to water was 5.5:1 and the concentration of Cd^2+^ was 100 mg/L, the saturation adsorption (*S_m_*) of soils 1 and 2 was 537.03 mg/kg and 547.49 mg/kg, respectively.

### 3.4. Effect of Solution pH Value on Cadmium Adsorption in Two Soils

The adsorption behavior of cadmium in the two soils with different solution pH values is shown in Figure 7 (cadmium solution concentration was 100 mg/L, water to soil ratio was 5.5:1). In general, the adsorption capacity increased as the pH became more alkaline. When the pH of the cadmium solution was 3.5, the cadmium adsorption capacity was significantly lower compared with the other three pH values; while at a pH value of 9.5, the adsorption capacity was significantly higher. When the pH of the cadmium solution was 4.5 and 5.5, the cadmium adsorption capacity was similar. The dynamic exponential asymptote equation was adopted to fit the adsorption kinetics of cadmium, as shown in Table 3: when the reaction time t tended to infinity, the saturation adsorption of *S_m_* in the two soils increased as the overall pH of the solution increased (the saturation adsorption of soils 1 and 2 was 536.27~540.7 mg/kg and 545.32~547.48 mg/kg, respectively). Under the same pH conditions, the saturation adsorption *S_m_* of soil 2 was greater than that of soil 1. In addition, when the reaction time was 24 h, the adsorption ratio of soils with different pH values was different (Figure 7C).

These results indicate that pH value may affect cadmium adsorption mechanisms in soils. Davis and Leckie (1978) [43] compared the intrinsic surface complexation constants of free metal ions and hydroxy-oxygen complexes on different metal oxides, and found that hydroxy-oxygen complexes are more stable. Firstly, under neutral and weakly alkaline conditions, the exchange affinity of Cd(OH)^+^ generated by the hydrolysis of heavy metal ions is greater than that of metal free Cd^2+^ ions, resulting in an increase in the adsorption ratio. Secondly, the increase of soil pH increases the negative charge capacity on the soil colloidal surface, which leads to the enhancement of the soil electrostatic action and, consequently, to the enhancement of the obligate adsorption capacity of the soil colloidal surface to metal ions. The mobility of cadmium is very low under neutral and basic pH conditions due to its precipitation as carbonate (otavite: CdCO_3_) and silicate (CdSiO_3_) minerals, as well as to its adsorption of ferric hydroxide/oxide and clay minerals [44]. However, the lower the pH value, the more H^+^ is present in the solution, leading to a greater competition between Cd^2+^ and H^+^ for adsorption sites, which results in a decrease of cadmium adsorption capacity in low-pH soils.

### 3.5. Factors Affecting Cadmium Adsorption in Undisturbed Soil Columns

According to previous studies, the cadmium adsorption capacity of marine sedimentary soils is greater than that of terrestrial sedimentary soils, and the adsorption rate is also faster. In an environment where intense interaction occurs between land and sea, the physical and chemical conditions of environmental media change dramatically, and the activity of heavy metals in sediments also changes correspondingly. Dou et al. (2015) [36] summarized the characteristics of elements in marine and terrestrial sediments based on multi-objective geochemical surveys, and found that the content of most elements was higher in the marine sediment than in the continental one.

In this paper, an analysis of the influence of soil pH, SOM and soil mechanical composition on cadmium adsorption was conducted on 18 groups of deep soil samples in order to reveal the key factors affecting cadmium adsorption in undisturbed soil columns.

#### 3.5.1. Effect of Soil pH on Cadmium Migration in Undisturbed Soil Columns

The above-mentioned studies on the effect of solution pH on cadmium adsorption in different soils suggest that pH is an important factor affecting cadmium adsorption capacity in both the terrestrial and marine sediments of the Nansha Area, and an increase in pH can enhance Cd^2+^ adsorption capacity. Figure 8 plots the variation of cadmium content and pH in the deep soil profile of the Nansha Area. The results showed that, with increasing soil depth, pH tended to increase overall, while cadmium content gradually decreased, which is exactly the opposite of what has been observed in the previous analysis. What is the reason for such a different result?

Previous studies [27] have shown that the surface soil in the Nansha area of the Pearl River Delta is to some extent polluted by cadmium, mainly because of anthropogenic inputs. Cadmium is mainly concentrated in the surface and analysis of pH and cadmium content in the 0–20 cm surface layer—which consists of silty clay—showed that soil pH was positively correlated with cadmium content (*p* < 0.01, correlation coefficient r = 0.554). The influence of soil pH on cadmium adsorption-desorption processes is not only reflected in the surface soil, but also in the longitudinal migration and transformation of this element. Cadmium content significantly decreases with depth, and its variation can be subdivided into three categories (Figure 9) based on pH differences: (1) between 0~20 cm of soil with a pH of 6.5~7.5, the soil is neutral and reaches the maximum average cadmium content. Between 20~50 cm of soil with pH > 7.5 (an increasing), cadmium content is significantly reduced until reaching stability. Figure 9A shows that the surface neutral soil inhibits the longitudinal migration of cadmium along the soil column; (2) between 0~20 cm of soil with a pH value of 5.5~6.5, the soil was weakly acidic, and the cadmium content reached its maximum. Between 20~50 cm and 50~90 cm the soil was neutral, and at a depth of more than 90 cm it became alkaline. The cadmium content at 20~50 cm slowly decreased with increasing depth, showing an initially slower rate of curve variation compared with the first soil category, and then a faster rate as depth continued to increase. Figure 9B shows that the degree of vertical migration of cadmium was slightly higher than in the first soil category; (3) between 0~20 cm of soil with a pH value of 4.5~5.5, the soil was acidic and the cadmium content was low. Between 20~50 cm and 50~90 cm the soil was weakly acidic: the average cadmium content in the first 20–50 cm layer reached its maximum, while the average cadmium content in the 50–90 cm layer decreased, though remaining still higher than that in the surface soil. The soil at a depth > 90 cm was neutral, and the content of cadmium continued to decrease with increasing depth (Figure 9C), indicating that the adsorption capacity in the surface acidic soil was poor, and that the longitudinal migration capacity of cadmium increased significantly with depth.

In general, the surface soil is neutral and alkaline, which can inhibit the longitudinal migration of cadmium. This is because in a weakly alkaline environment with a high pH, an increase in soil pH will determine the transition of cadmium into its carbonate state and Fe-Mn oxidation state, forming precipitation complexes, increasing the negative charge, and therefore reducing the mobility of all heavy metals [45]. In addition, the pH increase causes the OH^−^ in the solution to react with the groups on the surface of soil particles, resulting in the accumulation of negative charges; the smaller the competition of Cd^2+^ for the adsorption sites, the stronger the Cd^2+^ adsorption capacity of the soil, and the faster the adsorption rate. In contrast, in a weakly acidic environment, the cadmium adsorption in surface soil is limited, because the existing competitive adsorption of H^+^ and Cd^2+^ favors H^+^, as H^+^ ions can replace Cd^2+^ ions on the surface of the clay. This is due to the strong combination of H^+^ ions with oxygen atoms at the edge of the crystal [46]. In an acidic environment, Cd^2+^ could be effectively desorbed, and the lower the pH value is, the higher the reversibility of the adsorption process, as it is more difficult for Cd^2+^ to be adsorbed on the surface of soil particles. However, this will increase the extent of the longitudinal migration, allowing cadmium ions to migrate into deep soil layers and pollute phreatic aquifers.

#### 3.5.2. Soil Mechanical Composition

Based on the soil type, the deep profile soil in the Nansha area can be divided into two categories: (1) continental sediment, at depths of 0~20 cm, 20~50 cm, and 50~90 cm, where the soil type is silty clay; here, the clay content in the soil is higher, accounting for 30%~44.2%, and the soil adsorption capacity is strong. The clay content decreased slightly from top to bottom, and cadmium content at each depth was 0.39, 0.28, and 0.22 mg/kg, respectively; (2) marine sediment, at depths of 90–140 and 140–200 cm, where the soil type is mainly silty sand, with low clay content, weak soil adsorption capacity, and cadmium content at each depth ranging between 0.23–0.52 and 0.2–0.45 mg/kg, respectively (Figure 10). There was no significant positive correlation between clay content and cadmium adsorption capacity in both the terrestrial and marine sedimentary soils. However, in general, the longitudinal variation of cadmium content correlates well with the change of sediment grain size. In particular, the cadmium content in coarse-grained sediments prevalent in sandy soils was low, while the element was enriched in fine-grained sediments which mainly characterize cohesive soils. This conclusion is consistent with the research by Covelo et al. (2004) [47], which showed that soils with the highest heavy metal adsorption capacity also contained the highest contents of SOM, oxides, and clays. The specific surface area of soil clay particles is large, and the surface is usually negatively charged, resulting in a high cation exchange capacity. The positively charged cations will adsorb on the surface of clay minerals under the action of electrostatic force. Therefore, it is believed that the higher the content of physical clay particles in the soil, the stronger the ability to adsorb Cd^2+^.

Basic chemical properties of the deep soil measured in the study area, such as pH, CEC, SOM, and others, were correlated with the content of clay, silt, and sand in soil mechanical composition analysis. The proportion of clay was negatively correlated with soil pH (correlation coefficient = −0.684), and was strongly correlated with CEC and SOM (correlation coefficients were 0.893 and 0.776, respectively). The silt proportion was negatively correlated with soil pH (correlation coefficient = −0.647), and was strongly correlated with CEC and SOM (correlation coefficients were 0.867 and 0.861, respectively). The mechanical composition of soil at the sea-land transition zone affects its basic chemical properties, and consequently it affects the migration and transformation of cadmium.

#### 3.5.3. Soil Organic Matter

SOM is an important component of soil. The SOM content at 0~20, 20~50, and 50~90 cm depths was similar, with mean values of 19.2, 18.4, and 19.3 mg/kg, respectively. Similarly, the mean SOM content value at 90~140 and 140~200 cm depths was 15.9 and 13.6 mg/kg, respectively. As shown in Figure 5B, cadmium concentrations generally declined with the decrease of SOM content. Ren et al. (2020) [48] pointed out that after the removal of SOM, the cadmium adsorption capacity generally decreased. SOM has a large surface area and abundant oxygen-containing functional groups (carboxyl, phenolic, hydroxyl, etc.), which produce organic ligands through decomposition (organic acids and humus, etc.). A variety of organic acids (oxalic, citric, formic, acetic, malic, succinic, lactic, and fumaric acids) were observed to complexate with Cd^2+^ in the soil through ion exchange mechanisms, and to form a complex with a stronger stability than the exchange reaction [49], which may also reduce the mobility of heavy metals through precipitation [50]. As a result, the content of exchangeable and soluble cadmium in the soil can be effectively reduced, thus limiting the bioavailability of this element and the uptake potential of plants [51]. Therefore, the higher the SOM content is, the greater the adsorption capacity of the soil [50].

## 4. Conclusions

Soils in the Nansha area of the Pearl River Delta are seriously polluted, and cadmium concentration in the two sedimentary environments analyzed was significantly higher than in Guangdong Province (0.056 mg/kg) and the greater Pearl River Delta area (0.11 mg/kg). The marine sedimentary soil is alkaline and it has a greater colloidal surface negative charge, which enhances its electrostatic action, resulting in the enhancement of specific metal ion adsorption. SOM content in the terrestrial sedimentary soil is high, so organic and inorganic colloids combine to form new colloid compounds during the adsorption process, which weakens the adsorption capacity of soil particles to heavy metals. The adsorption characteristics of cadmium in the two soils can be described by the Henry, Langmuir, and Freundlich isotherm models. According to the fitting square variance, the Freundlich equation showed a better fitting performance. The fitting results showed that the isothermal cadmium adsorption constant *K_F_* in the marine sedimentary soil was significantly higher than that in the terrestrial sedimentary soil, measuring 558.21 and 363.04, respectively, and indicating that the Cd^2+^ adsorption capacity of the marine soil was stronger. The continental sedimentary soil, however, presented a high content of clay—the clay-specific surface area is large—and was usually negatively charged on the surface. This type of soil has the very high cation exchange capacity and, under the action of electrostatic force, positively charged cations can determine adsorption to the surface of clay minerals. As a result, the saturation adsorption of cadmium was higher in the terrestrial (2419.83 mg/kg) than in the marine sedimentary soil (2228.25 mg/kg). In addition, it was observed that when the surface soil in the study area was neutral and alkaline, the vertical migration of cadmium was inhibited; when it was weakly acidic, the cadmium adsorption capacity was limited and when the topsoil was acidic, Cd^2+^ was effectively desorbed. In summary, the lower the pH value is, the higher the reversibility of the adsorption process, and the more difficult it is for Cd^2+^ to be adsorbed to the surface of soil particles. This favors the longitudinal migration of cadmium through the soil column allowing cadmium ions to reach the deep soil and pollute phreatic aquifers. Overall, the results of this study showed that the distribution of cadmium in terrestrial and marine soils in the Nansha area of the Pearl River Delta is affected by adsorption and pH values.

## Figures and Tables

**Figure 1 ijerph-19-00793-f001:**
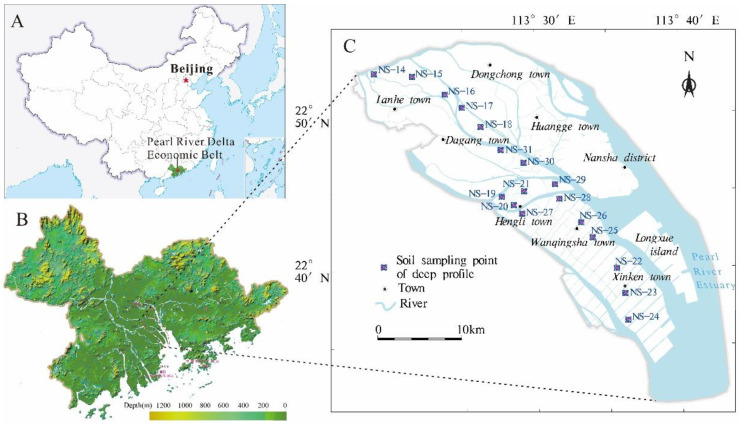
The location (**A**), terrain (**B**), and deep profile soil sampling maps (**C**) of the study area.

**Figure 2 ijerph-19-00793-f002:**
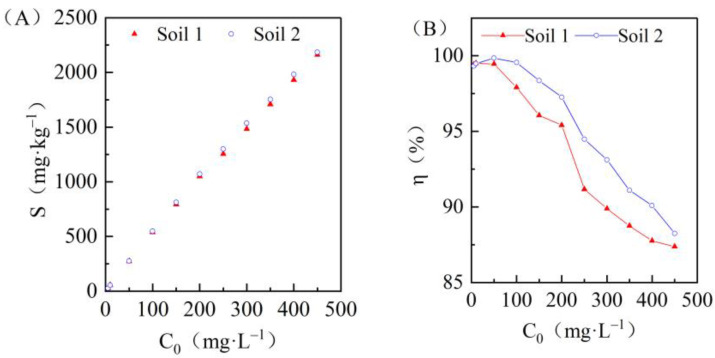
Changes in adsorption capacity *S* (**A**) and adsorption rate *η* (**B**) with the initial concentration of solution *C*_0_ in the two soils.

**Figure 3 ijerph-19-00793-f003:**
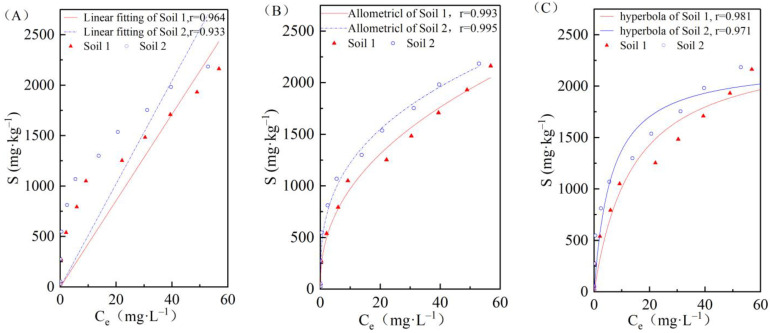
Cadmium adsorption isothermal curves for the two soils are shown in (**A**) Henry equation; (**B**) Freundlich equation; and (**C**) Langmiur equation.

**Figure 4 ijerph-19-00793-f004:**
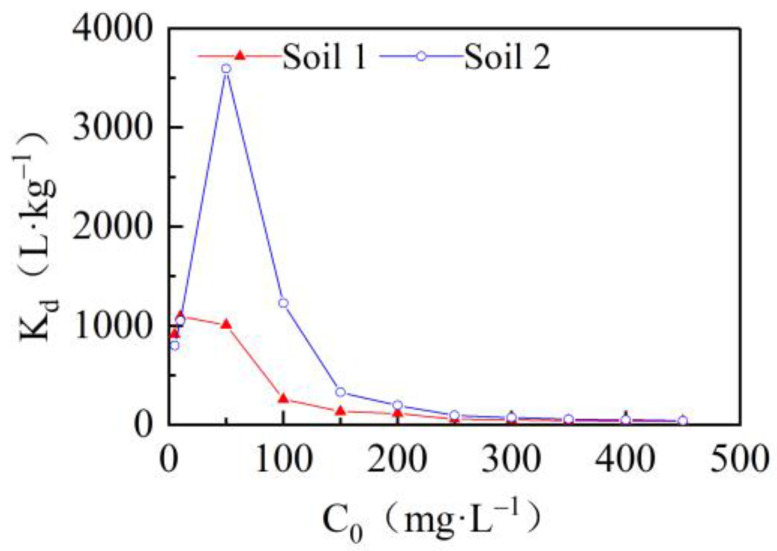
Relationship between the solid/liquid distribution coefficient *K_d_* and the initial concentration of cadmium *C*_0_ in two soils.

**Figure 5 ijerph-19-00793-f005:**
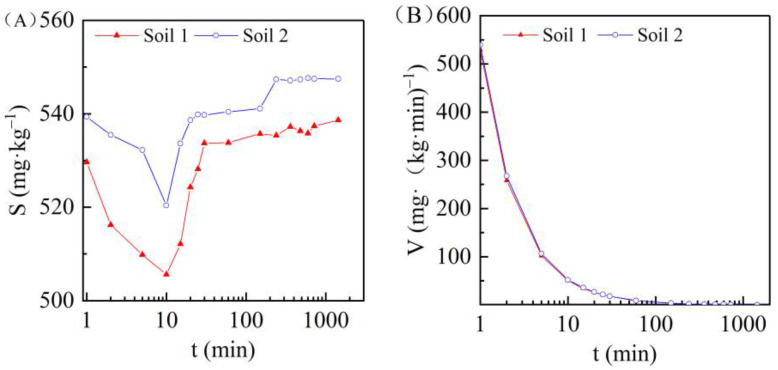
The oscillation time *t* affected the cadmium adsorption capacity *S* (**A**) and adsorption rate *V* (**B**) of the two soils.

**Figure 6 ijerph-19-00793-f006:**
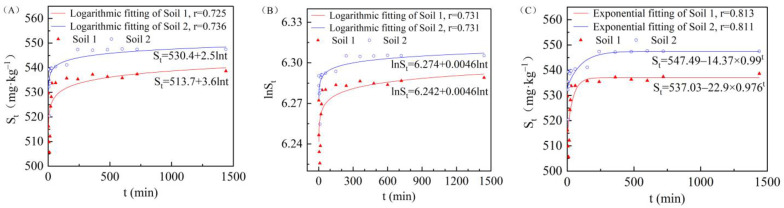
Elvoich equation fitting (**A**), biconstant equation fitting (**B**), and similar exponential asymptote equation fitting (**C**) for cadmium adsorption in the two soil types.

**Figure 7 ijerph-19-00793-f007:**
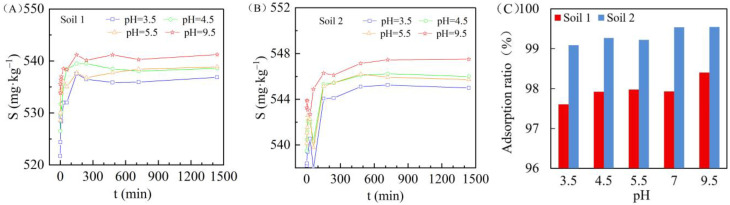
Relationship between oscillation time and cadmium adsorption capacity in soil 1 (**A**) and soil 2 (**B**) under different pH values, and adsorption ratio under different pH values in different soils (**C**).

**Figure 8 ijerph-19-00793-f008:**
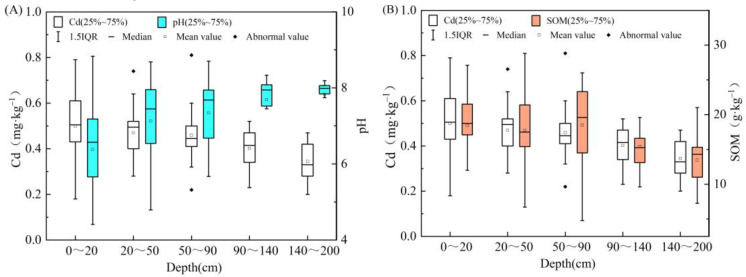
Box plots of soil cadmium content -pH (**A**) and cadmium content -SOM (**B**) at different depths along the deep soil profile.

**Figure 9 ijerph-19-00793-f009:**
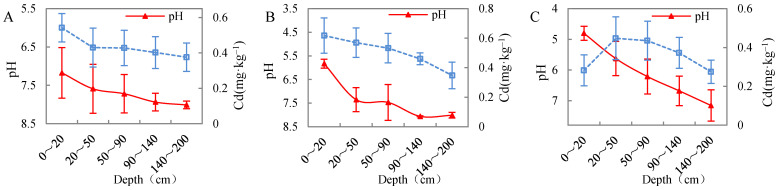
Comparison of three trends of cadmium content and pH in the deep soil profile (the error bar represents the standard deviation), between 0~20 cm of soil with a pH of 6.5~7.5 (**A**), a pH value of 5.5~6.5 (**B**), and a pH value of 4.5~5.5 (**C**).

**Figure 10 ijerph-19-00793-f010:**
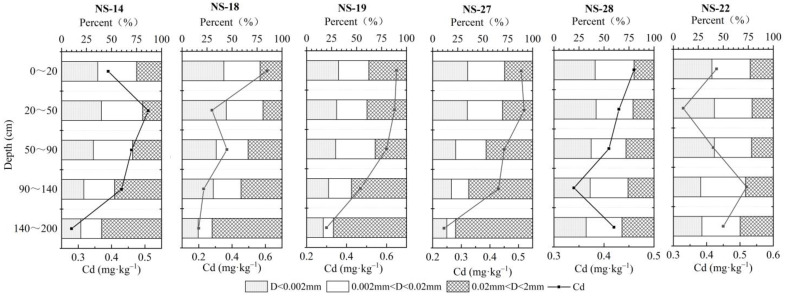
Changes in soil mechanical composition and cadmium content with depth in the deep profile soil (0–200 cm).

**Table 1 ijerph-19-00793-t001:** Basic physical and chemical properties of soils in two sedimentary environments.

	Soil Mechanical Composition Ratio (%)			SOM	CEC	pH	Cd
	D < 0.002 mm	0.002 mm < D < 0.02 mm	0.02 mm < D < 2 mm	g/kg	cmol/kg		mg/kg
Soil 1	35	36.8	28.2	18.0	15.2	6.40	0.52
Soil 2	18.6	17.4	64	15.8	10.5	8.08	0.48

**Table 2 ijerph-19-00793-t002:** Isothermal adsorption parameters of cadmium in the two soils.

Adsorption Model	Parameter	Unit	Soil 1	Soil 2
Henry equation	*K_d_*	L/kg	42.81	51.11
Freundlich equation	*K_F_*	-	363.04	558.21
	1/*n*	-	0.43	0.34
Langmiur equation	*K_L_*	-	0.072	0.162
	*S_m_*	mg/kg	2419.83	2228.25

**Table 3 ijerph-19-00793-t003:** Dynamic fitting of the exponential asymptote equation for the two soils under different solution pH values.

Soil	pH	Expression	COD (R^2^)	S_m_ (mg/kg)
Soil 1	pH = 3.5	S = 536.27 − 12.13 × 0.968 ^t^	0.899	536.27
	pH = 4.5	S = 538.85 − 11.11 × 0.959 ^t^	0.948	538.85
	pH = 5.5	S = 537.18 − 9.69 × 0.852 ^t^	0.864	537.18
	pH = 9.5	S = 540.70 − 5.64 × 0.977 ^t^	0.906	540.70
Soil 2	pH = 3.5	S = 545.32 − 6.84 × 0.933 ^t^	0.883	545.32
	pH = 4.5	S = 546.27 − 5.83 × 0.933 ^t^	0.875	546.27
	pH = 5.5	S = 546.14 − 5.24 × 0.994 ^t^	0.801	546.14
	pH = 9.5	S = 547.48 − 4.22 × 0.994 ^t^	0.915	547.48

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
