# Peer review of "Adsorption and pH Values Determine the Distribution of Cadmium in Terrestrial and Marine Soils in the Nansha Area, Pearl River Delta"

_ijerph, 2022, doi:10.3390/ijerph19020793_

Round 1
Reviewer 1 Report
Dear Authors,
The manuscript is correctly prepared and meets all the requirements related to the nature of a scientific publication. However, I have a few editorial remarks without which the text cannot be accepted for further publishing procedure.
The citations in the form of footnotes are superfluous; they can generally be in the text.
Please review the references again, reformatting them by the requirements of the editorial office of the journal. Moreover, look at the quoted publication from before 2010 - are they necessary? Is there no more recent data?
And the main thing - drawings. The quality of Figures 1, 2, 3 is unacceptable. The charts are complicated to read, which significantly reduces the value of the publication. (please remove the horizontal lines in Figure 6).
Yours faithfully
Author Response
Dear Reviewer:
On behalf of my co-authors, we thank you very much for giving us an opportunity to revise our manuscript, we appreciate Editor and Reviewers very much for their positive and constructive comments and suggestions on our manuscript entitled “Adsorption and pH values determine the distribution of cad-mium in terrestrial and marine soils in the Nansha area, Pearl River Delta”. (ID: ijerph-1502310).
Point 1: The citations in the form of footnotes are superfluous; they can generally be in the text.
Response 1: We looked up the previous papers in this journal and found that the references were usually cited in the form of footnotes, so we still cited the references in the form of footnotes in the main text.
Point 2: Please review the references again, reformatting them by the requirements of the editorial office of the journal. Moreover, look at the quoted publication from before 2010 - are they necessary? Is there no more recent data?
Response 2: Thank you very much for the reviewer's reminding. We reviewed the references again, retype set them, and deleted some references before 2010.
Point 3: And the main thing - drawings. The quality of Figures 1, 2, 3 is unacceptable. The charts are complicated to read, which significantly reduces the value of the publication. (Please remove the horizontal lines in Figure 6).
Response 3: The horizontal lines in Figure 6 have been removed (modified to Figure 9). We have adjusted the size of fonts and symbols in Figure 2 to Figure 6 to improve the clarity of the pictures, and submitted a clear JPG file for each picture in the attachment.
Reviewer 2 Report
-The authors should rewrite the abstract of the manuscript so that it is more in accordance with the work carried out. For example, the reference to Eh in the properties of soils is not the subject of the abstract of this work. -In section 2.1 the authors mention data on the climate of the study area, but do not include the source from which they extract this information. -Figure 1 must be cited prior to its appearance in the text. The authors refer only to Fig. 1.C and later. -Authors should include the specific characteristics of the equipment used on lines 146-150 instead of lines 173-175. -The authors could justify why they have been used by the equations of Henry, Langmiur and Freundlich in the study of adsorption isotherms, and the models of the Freundlich equation, double constant and pseudo-first-order dynamics in the study of the kinetics of adsorption?. -The experimentation expressed in section 3.2 has already been described previously in section 2 Figure 2.F is not named in the figure caption -In the bibliography section, the authors should include the numbering of the different citations to be easily identified when reading the text.Author Response
Dear Reviewer:
On behalf of my co-authors, we thank you very much for giving us an opportunity to revise our manuscript, we appreciate Editor and Reviewers very much for their positive and constructive comments and suggestions on our manuscript entitled “Adsorption and pH values determine the distribution of cad-mium in terrestrial and marine soils in the Nansha area, Pearl River Delta”. (ID: ijerph-1502310).
Point 1:The authors should rewrite the abstract of the manuscript so that it is more in accordance with the work carried out. For example, the reference to Eh in the properties of soils is not the subject of the abstract of this work.
Response 1: We revised the abstract of the manuscript and removed the content of the abstract that was not very relevant to the topic of the manuscript study.
Point 2: In section 2.1 the authors mention data on the climate of the study area, but do not include the source from which they extract this information.
Response 2: The sources of climate data and other information in the study area are supplemented in the form of references on line 117.
Point 3:Figure 1 must be cited prior to its appearance in the text. The authors refer only to Fig. 1.C and later.
Response 3: Fig.1 has been added on line 114.
Point 4: Authors should include the specific characteristics of the equipment used on lines 146-150 instead of lines 173-175.
Response 4: Lines 144-147 test parameters such as soil pH value and organic matter, while lines 170-172 test pH value and conductivity of solution, so the instruments are different. The test method is added on lines 144-147.
Point 5: The authors could justify why they have been used by the equations of Henry, Langmiur and Freundlich in the study of adsorption isotherms, and the models of the Freundlich equation, double constant and pseudo-first-order dynamics in the study of the kinetics of adsorption?
Response 5: We have explained the use of the Henry, Langmiur, and Freundlich equations in adsorption isotherm studies, and the use of the Freundlich equation, double constant and pseudo-first-order dynamics in adsorption kinetics studies on Lines 203-206 and 217-221 respectively.
Point 6: The experimentation expressed in section 3.2 has already been described previously in section 2 Figure 2.F is not named in the figure caption.
Response 6: Section 2 introduces the operation method and data analysis method of the experiment. Section 3.2 and Figure 2 (modified to Figure 2, 3 and 4) are the results and discussion of the experiment. Fig.2F modified to Fig.4.
Point 7: In the bibliography section, the authors should include the numbering of the different citations to be easily identified when reading the text.
Response 7: The references have been numbered as required by the reviewer.
Reviewer 3 Report
A well planned work, well conducted. the results seems reliable and the conclusions are sound. The paper can be accepted in the present form.
Author Response
Dear Reviewer:
On behalf of my co-authors, we thank you very much for giving us an opportunity to revise our manuscript, we appreciate Editor and Reviewers very much for their positive and constructive comments and suggestions on our manuscript entitled “Adsorption and pH values determine the distribution of cad-mium in terrestrial and marine soils in the Nansha area, Pearl River Delta”. (ID: ijerph-1502310).
Point 1: A well planned work, well conducted. the results seems reliable and the conclusions are sound. The paper can be accepted in the present form.
Response 1: Thank you very much for the reviewers' affirmation of the manuscript. We have made improvements in the above areas.
We tried our best to improve the manuscript and made some changes in the manuscript. We would like to express our great appreciation to you for comments on our paper, and hope that the correction will meet with approval.
Thank you and best regards.